**Title:**
**SOIL CARBON DIOXIDE EMISSIONS CONTROLLED BY AN EXTRACELLULAR**
**OXIDATIVE METABOLISM IDENTIFIABLE BY ITS ISOTOPE SIGNATURE.**
**Authors:**
**B. Kéraval [1,2,3], A .-C. Lehours[1,2], J. Colombet[1,2],  C. Amblard [1,2], G. Alvarez [3,4], S. Fontaine [3]**
**Authors affiliations**
[1] Clermont Université, Université Blaise Pascal, Laboratoire Microorganismes : Génome et
Environnement, BP 10448, 63000, Clermont-Ferrand, France
[2] CNRS, UMR 6023, Laboratoire Microorganismes : Génome et Environnement, 63178 Aubière, France
[3] INRA, UR874 (Unité de Recherche sur l'Ecosystème Prairial), 5 Chemin de Beaulieu, 63039 Clermont-
Ferrand, France.
[4] Clermont Université, VetAgro Sup, BP 10448, F-6300 Clermont-Ferrand, France
Correspondence to: B. Kéraval (benoit.keraval@gmail.com)

**ABSTRACT**

Soil heterotrophic respiration is a major determinant of carbon (C) cycle and its interactions with climate. Given the complexity of the respiratory machinery it is traditionally considered that oxidation of organic C into carbon dioxide ($CO_2$) strictly results from intracellular metabolic processes. Here we show that C mineralization can operate in soils deprived of all observable cellular forms. Moreover, the process responsible of $CO_2$ emissions in sterilized soils induced a strong C isotope fractionation (up to 50 ‰) incompatible with a respiration of cellular origin. The supply of [13]C-glucose in sterilized soil led to the release of [13]$CO_2$ suggesting the presence of respiratory-like metabolism (glycolysis, decarboxylation reaction, chain of electron transfer) carried out by soil-stabilized enzymes and by soil mineral and metal catalysts. These findings indicate that $CO_2$ emissions from soils can have two origins: 1) the well-known respiration of soil heterotrophic microorganisms and 2) an extracellular oxidative metabolism (EXOMET) or, at least, catabolism. These two metabolisms should be considered separately when studying effects of environmental factors on the C cycle because they do not likely obey the same laws and respond differently to abiotic factors.

**INTRODUCTION**

Mineralization of soil organic matter (SOM) into $CO_2$ and mineral nutrients is central to the functioning of eco- and agro-systems in sustaining nutrient supply and plant primary production. Soil carbon (C) mineralization is also a major determinant of the global C cycle and climate by releasing from land surfaces an equivalent of ten times the anthropogenic emissions of $CO_2$ (IPCC, 2007; Paterson and Sim, 2013). Therefore, knowledge of the metabolic pathways by which SOM is oxidized is crucial to predicting both the food production and the climate under a changing environment.

It is traditionally considered that SOM mineralization result from the activity of soil microbial communities through biological catalyzed processes including both extracellular depolymerization and cellular metabolisms. Extracellular depolymerization converts high-molecular weight polymers like cellulose into soluble substrates assimilable by microbial cells. This depolymerization is performed by extracellular enzymes released in soil through microbial cell excretion and lysis (Burns et al., 2013). In cells, assimilated substrates are carried out by a cascade of endoenzymes (Sinsabaugh et al., 2009; Sinsabaugh and Follstad Shah, 2012), along which protons and electrons are transferred from a substrate to intermediate acceptors (*e.g.* NADP) and small C compounds are decarboxylated into $CO_2$. At the end of the cascade, the final acceptor (*e.g.* $O_2$ under aerobic conditions) receives the protons and electrons while the gradient of $H^+$ generated is used by ATP-synthase to produce ATP (Junge et al., 1997).

Given the complexity of its machinery it is often believed that respiration is strictly an intracellular metabolic process. However, this paradigm is challenged by recurrent observations of persistent substantial $CO_2$ emissions in soil microcosms where sterilization treatments (*e.g.* γ-irradiations) reduced microbial activities to undetectable levels (Blankinship et al., 2014; Kemmitt et al., 2008; Lensi et al., 1991; Maire et al., 2013; Ramsay and Bawden, 1983; Trevors, 1996). Maire *et al.* (2013) addressed this issue and proposed that extracellular oxidative metabolisms (EXOMET) contribute to soil respiration. According to these authors, intracellular enzymes involved in cell oxidative metabolism are released during cell lysis and retain their activities in soil thanks to the protective role of soil particles. These enzymes are able to oxidize [13]C-glucose to [13]$CO_2$ using $O_2$ as the final electron acceptor suggesting that all or part of the cascade of biochemical reactions involved in cell oxidative metabolism are reconstructed outside the cell (Maire et al., 2013). As an alternative explanation Blankinship *et al.* (2014) proposed that some decarboxylases, retaining activities outside the cell in sterilized soils, catalyze $CO_2$ emissions through decarboxylation of intermediary metabolites of the Krebs cycle. Whereas differing in the complexity of the proposed mechanisms, these

results (i) suggest that $CO_2$ emissions from soils are not only dependent to the bio-physicochemical
environment provided by the cells, (ii) indicate that the soil micro-environment heterogeneity offers a range
of physicochemical conditions allowing endoenzymes to be functional.
Despite these recent advances, the paradigm that only a cell can organize the complex machinery achieving
the complete oxidation of organic matter, at ambient temperature, remains established in the scientific
community (see published discussions generated by Maire *et al.,* 2012). In this vein, some authors suggested
that $CO_2$ emissions from γ-irradiated soils can result from "ghost cells" (non-proliferating but
morphologically intact cells) which conserve some cellular metabolic activities during prolonged periods of
time (Lensi et al., 1991; Ramsay and Bawden, 1983).
The objective of the present study was to determine whether a purely extracellular oxidative metabolism
(EXOMET) can occur in a soil deprived of active and "ghost" cells. To this aim, high doses of γ-irradiations
and different time of soil autoclaving were combined to suppress both biomass and necromass ("ghost"
cells). The presence/absence of active and non-active cells in soil was checked by observations with
transmission electron microscopy on tangential ultrathin sections of soil, DNA and RNA soil content and
flow cytometry. The production and the isotope composition ($\delta^{13}C$) of $CO_2$ were monitored in sterilized and
non-sterilized soils over 4 periods through 91 days of incubation. We also tested whether the EXOMET in
sterilized soils can carry out complex cascade of biochemical reactions (e.g. an equivalent of glycolysis and
Krebs cycle) by incorporating $^{13}C$- labelled glucose and by quantifying emissions of $^{13}C\text{-}CO_2$ **(Fig 1)**.

## 83 MATERIAL AND METHODS

### 84 Soil sampling, sterilization and incubation

Samples were collected in November 2012 from the 40-60 cm soil layer at the site of Theix (Massif Central,
France). The soil is sandy loam Cambisol developed on granitic rock (pH=6.5, carbon content = 23,9±1 g
C kg$^{-1}$). For detailed information on the site see Fontaine *et al*. (Fontaine et al., 2007)**.** Fresh soil samples
were mixed, sieved at 2 mm, dried to 10 % and irradiated with gamma ray at 45 kGy ($^{60}Co$, IONISOS,
ISO14001, France). To demonstrate the absence of viable cells in soil after irradiation, we inoculated culture
medium for bacteria (LB agar) and fungi (Yeast Malt agar) with irradiated soil and we applied CARD-FISH
to irradiated soil extracts. Results showed the absence of any microbial proliferation and RNA-producing
cells (Maire et al., 2013). After irradiation, some sets of soil samples were exposed to autoclaving at 121°C
during variable periods (0.5 h, 1 h, 1.5 h, 2 h, 4 h). Incubated microcosms consisted of 9 g (oven dried basis)
samples of sieved soils placed in 120 mL sterile glass flasks capped with butyl rubber stoppers and sealed
with aluminum crimps. Microcosms were flushed with a sterilized free $CO_2$ gas (80 % $N_2$, 20 % $O_2$) and
incubated in the dark at 20°C for 91 days. Non-irradiated living soil was also incubated as a control. Three
microcosm replicates per treatment were prepared. Flasks were sampled at 15, 31, 51 and 91 days of
incubation to measure $CO_2$ fluxes and $^{13}C$ abundance of $CO_2$. After each measurement, flasks containing
soil samples were flushed with a sterilized free $CO_2$ gas (80 % $N_2$, 20 % $O_2$). All manipulations were done
under sterile conditions. In the text and the figures LS mean "living soils", IS mean "irradiated soils" and
IAS-t referred to irradiated and autoclaved soils with 't' referring to the time of autoclaving.

### 103 Carbon dioxide emissions and their isotope composition ($^{13}C/^{12}C$)

The amount and isotope composition ($\delta^{13}C$) of $CO_2$ accumulated in flasks during the incubation period
were quantified using a cavity ring down spectrometer analyser coupled to a small sample injection module
(Picarro 2101-i analyser coupled to the SSIM, Picarro Inc., Santa Clara, CA, USA). A volume of 20 ml of
gas was sampled by the analyser. The $CO_2$ concentration in gas samples ranged from 300 to 2000 ppm of
$CO_2$ in accordance with the operating range of the analyser. The $CO_2$ concentrations and delta $^{13}C$ of gas
samples were measured at a frequency of 0.5 Hz during 10 min. Value provided by the analyser is the
integrated value during these 10 min of measurement. A reference gas with a known concentration of $CO_2$
and delta $^{13}C$ was injected between samples. For each period of incubation, the cumulated amount of $CO_2$
was divided by the duration of the period (in days) to estimate the mean daily $CO_2$ emission rate.

**Content and isotope composition of dissolved organic carbon (DOC)**

At the beginning and at the end of the incubation (t = 15 and t= 91 days), DOC was extracted from 5 g of
soil with a 30 mM $K_2SO_4$ solution. After filtration through 1.6 µm (GE Healthcare, Life Sciences,
Whatman$^{TM}$, Glass microfiber filters), extracts were lyophilized. The lyophilized samples were analyzed
with an elementary analyzer (EA Carlo ERBA NC 1500) coupled to an Isotope Ratio Mass Spectrometer
(Thermo Finnigan DELTA S) to determine their carbon content and isotope composition (delta $^{13}C$).

**Isotope systematic**

We use standard δ notation for quantifying the isotopic composition of $CO_2$ and of DOC: the ratio R of
$^{13}C/^{12}C$ in the measured sample is expressed as a relative difference (denoted $\delta^{13}C$) from the Vienna Pee
Dee Belemnite (VPDB) international standard material. The carbon isotope composition is expressed in
parts per thousand (‰) according to the expression: $\delta^{13}C = (R_{sample}/ R_{VPDB}) - 1) \times 1000$. The carbon isotope
fractionation was calculated as follows: $\Delta\delta^{13}C$ (‰) $= (\delta^{13}C\text{-}DOC - \delta^{13}C\text{-}CO_2)/(1 + \delta^{13}C\text{-}CO_2)$.

**Soil cell density**

At the end of the incubation setting (t = 91 days), cells were separated from soil particles and enumerated
by flow cytometry (FC). One gram of soil was mixed with 10 mL of pyrophosphate buffer (PBS 1X, 0.01
M $Na_4P_2O_7$) and shaken for 30 min in ice at 70 rpm on a rotary shaker. After shaking, the solution was
sonicated 3 times (1 min each) in a water bath sonicator (Fisher Bioblock Scientific 88156, 320W, Illkirch,
France). Larger particles were removed by centrifugation (800 × g, 1 min); the supernatant was fixed with
paraformaldehyde (4 % final concentration) and stored at 4°C prior to quantification analysis. Total cells
counts were performed using a FACSCalibur flow cytometrer (BD Sciences, San Jose, CA, USA) equipped
with an air-cooled laser, providing 15 mW at 488 nm with the standard filter set-up. Samples were diluted
into 0.02 µm filtered TE buffer, stained with SYBR Green 1 (10,000 fold dilution of commercial stock,
Molecular Probes, Oregon, USA) and the mixture was incubated for 15 min in the dark. The cellular
abundance was determined on plots of side scatter versus green fluorescence (530 nm wave-length,
fluorescence channel 1 of the instrument). Each sample was analyzed for 1 min at a rate of 20µL.min$^{-1}$. FCM
list modes were analyzed using CellQuest Pro software (BD Biosciences, version 4.0). Cell density was
expressed as cells × g$^{-1}$ of soil (dry mass).

**Density and integrity of cells**

At the end of the incubation setting (t= 91 days), abundance of unicellular organisms (prokaryotic and
eukaryotic) with a preserved morphology was quantified on soil ultrathin sections (90 nm thick) by TEM.
Each step of the soil inclusion protocol was followed by centrifugation (12000 x g, 2 min) to pellet soil
samples. Aliquot of soil sample (0.05 g) was fixed for 1 hour in 1.5 mL of a Cacodylate buffer pH 7.4 (0.2
M cacodylate, 6 % glutaraldehyde and 0.15 % ruthenium red). Soil was washed three times with cacohydrate
0.1 M buffer during 10 min. Post fixation was conducted with the 0.1 M cacohydrate buffer containing 1 %
of osmic acid. To facilitate the further penetration of propylene oxide, soil dehydration was made through a
gradient of ethanol: 50 % ethanol (3 x 5 min), 70 % ethanol (3 x 15 min), 100 % ethanol (3 X 20 min)
solutions. To improve the resin permeation, the sample was incubated in a propylene oxide bath (3 x 20
min). To allow the sample to soak resin, soil sample was incubated overnight in a bath containing propylene
oxid and Epon 812 resin (ration 1:1), and secondary eliminated by flipping. After polymerization of cast
resin on soil preparations (48 h, 50°C), the narrower parts of the molded and impregnated aggregates were
pyramidally shaped with a Reichert TM60 ultramill and finally ultra-thin sections (90 nm) were performed
with a diamond knife (Ultra 45°, MF1845, DIATOME, Biel-Bienne, Switzerland; Ultramicrotome Ultracut
S, Reichert Jung Laica, Austria). Soil cuts were collected onto 400-mesh Cu electron microscopy grid
supported with carbon-coated Formvar film (Pelanne Instruments, Toulouse, France). Each grid was
negatively stained for 30 s with uranyl acetate (2 %), rinsed twice with 0.02 µm distilled water and dried on
a filter paper. Soil ultrathin sections were analyzed using a JEM 1200EX TEM (JEOL, Akishima, Japan).
Abundance of morphologically intact cells were expressed as cells x $mm^{-2}$ of soil.

## Soil DNA and RNA content

Two grams of soil were collected at the end of the incubation setting (t= 91 days). Genomic DNA and total
RNA were extracted from soil samples and purified using the PowerSoil DNA isolation kit and the
PowerSoil total-RNA isolation kit (Mo Bio Laboratories, Inc.), respectively. DNA and RNA content of soil
communities were visualized by electrophoresis on a 1 % agarose gel containing ethidium bromide (0.5
$g.mL^{-1}$) normalized with a 1 kbp size marker (Invitrogen). Negative control was performed as well.
Following electrophoresis, agarose gels were analyzed using ImageJ software (available at
http://imagej.nih.gov/ij/). The band intensities were used to quantify the relative content of soil DNA and
RNA in sterilized soils related to living soil.

## Soil incubations with $^{13}C_6$-labelled-glucose

Samples (9 g, dry mass basis) of irradiated (45 kGy) and autoclaved (121 °C, 4 h) soil were incubated after
addition of sterile solutions (1.53 mL of a 0.086 M glucose solution) of unlabelled- or of $^{13}C_6$- glucose ($^{13}C$
Abundance = 99 %). This amendment corresponds to 2.6 mg glucose $g^{-1}$ soil. Incubation and gas
measurements were performed as previously described.

## Statistical analyses

Each treatment was prepared in triplicate (n=3). One-Way ANOVA analysis was used to test the
involvement significance of sterilization treatments on $CO_2$ emissions, $\delta^{13}C$-$CO_2$, DOC, and $\delta^{13}C$-DOC.
Normality was tested using the Shapiro-Wilk test (p>0.05). Equality of variances were tested with a Leven's
Test (p<0.05). Student test analyses were used to test the significance of the difference (p<0.05) obtained
between each conditions. Those statistical analyses were performed using the PAST software V3.04
(Hammer, 2001).

**RESULTS**
**Effect of sterilization treatments**
**Microbial cell density and soil DNA and RNA content**
Gamma-irradiations did not significantly reduce cellular density as revealed by flow cytometry ($3.1 \times 10^8 \pm$
$1.3 \times 10^7$ cell.g$^{-1}$ in living soil, LS, *versus* $3.2 \times 10^8 \pm 1.1 \times 10^8$ cell.g$^{-1}$ in irradiated soil, IS, **Fig. 2a**) and
transmission electron microscopy ($1.4 \ 10^4 \pm 4.3 \ 10^3$ in LS *versus* $9.5 \ 10^3 \pm 0.7 \ 10^2$ cell.g$^{-1}$ in IS, **Figs. 2b**
**and 2c**). However, two proxies of cellular functionality and activity (DNA and RNA) were substantially
decreased by irradiations (-93.5 % $\pm$ 1 % for DNA and -74 % $\pm$ 6 % for RNA, **Figs. 2d and 2e**). Moreover,
RNA and DNA streaks observed on electrophoresis gels indicated that the nucleic acid content of irradiated
soils was largely degraded (data not shown).
The combination of $\gamma$-irradiations and autoclaving decreased cell densities by two orders of magnitude in
irradiated and autoclaved soil, IAS (**Fig. 2a**). Results from flow cytometry and transmission electron
microscopy showed that the cell density was reduced to < 2% compared to LS. After autoclaving,
transmission electron microscopy revealed that the cell density was reduced to undetectable values (**Figs.**
**2b**). According to transmission electron microscopy and nucleic acid extract results (**Figs. 2b, 2d and 2e**),
the remaining flow cytometry signal in IAS is attributed to auto fluorescent particles and unspecific binding
of the fluorescent dyes on debris.
**Dissolved organic carbon (DOC) and its isotopic composition**
Both $\gamma$-irradiations and autoclaving modified the soil chemistry as revealed by the analysis of the aqueous
phase at the beginning of the experiment. The aqueous phase contained much more DOC in irradiated soil
than in untreated soil ($37\pm3$ µg C.g$^{-1}$ to $303\pm17$ µg C.g$^{-1}$ in LS and IS, respectively (**Fig. 3a**). Autoclaving
further increased DOC content which gradually accumulated according to the time of autoclaving, from
$557\pm11$ µg C.g$^{-1}$ with 0.5 h of autoclaving to $1060\pm 28.4$ µg C.g$^{-1}$ after 4 h of autoclaving (**Fig. 3a**).
Similarly, the $\delta^{13}$C-DOC gradually increased from $-27.4 \pm 0.4$ ‰ in LS to $-24.9\pm 0.12$ ‰ in IAS-4h (**Fig.**
**3b**). In all soil microcosms, DOC content and $\delta^{13}$C of DOC did not significantly change over time (data not
shown).
All soil microcosms emitted $CO_2$ throughout the incubation (**Fig. 3c**). The daily $CO_2$ emission rate increased
significantly ($p < 0.05$) with time in LS whereas it gradually declined in IS (**Fig. 3c**). All IAS microcosms
exhibited similar dynamics of daily $CO_2$ emission rate: the high daily $CO_2$ emission rate recorded during
the first period of incubation (0-15 days) strongly decreased during the second period (15-31 days) and
stabilized thereafter (**Fig. 3c**).
Cumulated $CO_2$ emissions from LS and IS were not significantly ($p < 0.05$) different throughout the 91 days
of incubation ($24.4 \pm 1.5$ and $21.9 \pm 1.3$ µgC.g$^{-1}$ in LS and IS, respectively) but were significantly ($p < 0.05$)
higher than cumulated $CO_2$ emissions from IAS treatments ($16.8 \pm 1.5$ µgC.g$^{-1}$)(Data not shown). The
duration of autoclaving has no effect on cumulated $CO_2$ emissions. At the end of the incubation, the
percentage of initial DOC oxidized to $CO_2$ was low for all sterilized soils (< 7.2%) and decreased with the
duration of autoclaving (from 2.9 to 1.8% for  IAS 0.5H and IAS 4H, respectively)(See Supplement S1).
The $\delta^{13}C$-$CO_2$ from LS decreased with time, from -22.2 ± 0.1‰ to -28.9 ± 0.3‰. The $\delta^{13}C$-$CO_2$ strongly
decreased with the intensity of sterilization treatments, from -29.2 ± 1‰ in IS to -75.4 ± 2.8‰ in IAS with
4h of autoclaving (**Fig. 3d**). This pattern of values was maintained throughout the incubation but the
difference of $\delta^{13}C$-$CO_2$ between living and sterilized soils was maximal during the two intermediate periods
(P2 and P3).

**Carbon isotope fractionation during DOC mineralization**

The $\delta^{13}C$ strongly deviated between DOC and $CO_2$ in all sterilized soil microcosms (**Fig. 3e**) indicating
substantial C isotope fractionation during DOC mineralization. This isotope fractionation gradually
increased with the intensity of the autoclaving treatment, from 13.2 ± 0.7 ‰ in IAS with 0.5h of autoclaving
to 31 ± 2.5 ‰ in IAS with 4 h of autoclaving. The isotope fractionation was significantly and positively
correlated to the DOC content (r = 0.96, **Fig. 3e**). The $\delta^{13}C$ deviation between DOC and $CO_2$ in LS was <
4‰ (data not shown).

**Response of sterilized soil to supply of unlabelled and $^{13}C_6$ labelled glucose**

The supply of unlabelled or labelled glucose in IAS with 4h of autoclaving did not significantly change total
$CO_2$ emissions (data not shown). The $\delta^{13}C$ values of $CO_2$ released from microcosms with unlabelled glucose
ranged from -40.2 ± 0.6 ‰ to -53.8 ± 1.2 ‰ (**Fig. 4**). The $CO_2$ released from microcosms with $^{13}C$-glucose
showed progressive $^{13}C$ enrichment with time, from $\delta^{13}C$= 127.8 ± 1.3 ‰ to 657± 1.7 ‰ after 12 and 34
days of incubation, respectively (**Fig. 4**). At the end of the incubation, the amount of $^{13}C$-glucose released
as $CO_2$ corresponded to 0.01% of glucose input.

**DISCUSSION**

**Irradiation & autoclaving: an efficient combination to remove all traces of cell from soils.**

Demonstrating that complex soil matrices are truly devoid of intact cell is a challenging task. In previous
studies, measures for assessing abundance and activity of cells in γ-irradiated soils ranged from cultivation
(Blankinship et al., 2014; Maire et al., 2013), live-dead staining (Blankinship et al., 2014), fluorescent *in*
*situ* hybridization (Maire et al., 2013), biomass estimation (Maire et al., 2013), to biomarkers concentrations
(Buchan et al., 2012). All gave the same conclusion: a high proportion of dead but intact cells remained
after γ-irradiations of soil samples (Blankinship et al., 2014; Lensi et al., 1991; Maire et al., 2013). We found
a similar result using flow cytometry, transmission electron microscopy and estimation of DNA and RNA
content of soil (**Fig.2**).
To remove the remaining cells, we combined γ-irradiations with a time-gradient of autoclaving to analyze
the kinetics of microbial cellular lysis. To ensure that none cell with a preserved morphology remained in
soil aggregates we performed *in situ* observations with transmission electron microscopy on tangential
ultrathin sections of soil. This approach allows avoiding the pitfalls of methods involving dilute suspensions
of soil extracts (*i.e.* incomplete elution of microorganisms (Li et al., 2004). The combination of both
sterilization treatments allowed suppressing all observable cell structure (**Fig.2**). Our results also indicate
that the sterility of soil microcosms was maintained until the end of incubation.
By destroying the microbial biomass and releasing its content in soil, the sterilization treatments led to an
accumulation of DOC (**Fig.3a**). The increasing DOC accumulation with increasing time of autoclaving
likely resulted from desorption of organic carbon from soil particles (Berns et al., 2008) and/or from
depolymerization of carbohydrates (Tuominen et al., 1994) since microbial biomass was mostly lysed after
0.5h of autoclaving.

**Body of evidence for EXOMET**
The irradiated and autoclaved soils showed persistent (>91 days) and substantial soil $CO_2$ emissions (50-
80% of $CO_2$ emissions compared to LS). Those $CO_2$ emissions can hardly be ascribed to residual activities
of living and "ghost" cells since the sterilizing treatments removed all observable cell structure. Moreover,
the substantial C isotope fractionation (from 13 ‰ to 35 ‰, **Fig.3e**) induced by the process responsible of
$CO_2$ emissions is incompatible with a respiration of cellular origin. A substantial contribution of soil
carbonates to $CO_2$ emissions is unlikely because (i) the inorganic carbon pool is very small in the acidic soil
used in this study (Fontaine et al., 2007), (ii) the isotopic composition of $CO_2$ did not reflect the signature
of soil carbonates (Bertrand et al., 2007). The decarboxylation of organic compounds by a combustion
induced by sterilization treatments is also excluded because (i) $CO_2$ emissions were persistent throughout
the incubation, (ii) the C isotope fractionation during organic C combustion is typically weak (~3‰)
(Turney et al., 2006). Finally, irradiation and heating induce a heavy oxidative stress through the formation
of hydroperoxides, carboxyls and free radicals. These highly reactive oxidants can lead to organic matter
oxidation and decarboxylation. However, this oxidative process can hardly explain the persistent $CO_2$
emissions observed in our experiment since the half-life of highly reactive oxidants is extremely short (i.e.
$10^{-9}$ s for free radicals). Moreover, Blankinship *et al*. (2014) have shown that the persistence of soil $CO_2$
emissions after microbial biomass suppression (or at least reduction) is not specific to irradiated soil but
also occurs with other methods of sterilization such as chloroform fumigation and autoclaving.
The most parsimonious explanation of persistence of $CO_2$ emissions (**Fig. 3c**) and $O_2$ consumption (Maire
et al., 2013) after soil sterilization is an extracellular oxidative metabolism (EXOMET). By EXOMET we
suggest a cascade of chemical reactions where electrons are transferred from organic matter to redox
mediators (i.e. $NAD^+$/NADH, $Mn^{3+}$/$Mn^{2+}$) and finally to $O_2$. Those reactions can be catalyzed by respiratory
enzymes stabilized on soil particles (Maire et al., 2013) and by minerals and metals present in soil
(Blankinship et al., 2014; Majcher et al., 2000). The evidence of a complex oxidative metabolism is
supported by the oxidation of $^{13}$C-glucose to $^{13}CO_2$ (**Fig. 4**). Indeed, glucose is a stable molecule which must
undergo many biochemical transformations before being oxidized to carbon dioxide. The glucose
decarboxylation (**Fig. 4**) and concurrent $O_2$ consumption (Maire et al., 2013) suggest that EXOMET is able
to reconstitute an equivalent of glycolysis and Krebs cycle.
Mineral catalysts are stable and soil-stabilized enzymes are protected against denaturation (Carter et al.,
2007; Gianfreda and Ruggiero, 2006; Nannipieri, 2006; Nannipieri et al., 1996; Stursova and Sinsabaugh,
2008)**.** This stability of soil catalysts likely contributes to the maintenance of glucose oxidation and $CO_2$
emissions after soil exposure to high temperature and pressure (autoclaving). **Maire et al. (2013)** have
already pointed at the exceptional resistance of soil $CO_2$ emissions to high temperature, pressure and toxics.
However, by providing here the evidence of an oxidation of $^{13}$C-labelled glucose in γ-sterilized soil exposed
to high temperature and pressure, we show that the complex metabolic pathways of the EXOMET are
maintained under these extreme conditions.
**Origin of the C isotope fractionation during EXOMET**
Our results indicated that the EXOMET preferentially oxidizes organic molecules containing light ($^{12}$C)
over heavy ($^{13}$C) carbon atoms. Similar strong isotope fractionation has already been described during wet
abiotic oxidation of oxalic acid (Grey et al., 2006). The preferential conversion of substrate containing
lighter isotopes agrees with classical kinetic and thermodynamic laws. The presence of $^{13}$C atoms in a
substrate slows its conversion rate because of the higher activation energy request to induce the reaction
(Christensen and Nielsen, 2000; Heinzle et al., 2008). Classical works on thermodynamic also indicate that
the isotopic fractionation is dependent on substrate concentration (Agren et al., 1996; Goevert and Conrad,
2009; Wang et al., 2015). Under limited substrate concentration, the isotope fractionation decreases because
the heavy molecules left over during the first stages of reaction are finally carried out by the process.
Consistently, the isotopic fractionation induced by the EXOMET was quantified with an excess of substrates
(S1). Moreover, the magnitude of isotope fractionation was positively correlated to DOC content (**Fig. 2e**).
However, the causal link between the magnitude of fractionation and the DOC content is not certain since
the correlation emerges from a compilation of results obtained after different sterilization treatments. Further
studies should analyze this causal link in experiments where the DOC content is directly manipulated.
Previous studies (Blair et al., 1985; Zyakun et al., 2013) have shown that, contrary to EXOMET, cells
induced no or few ($< 4$‰) C isotope fractionation during respiration. This difference between cell respiration
and EXOMET can be explained by two processes. First, substrate absorption by microbial cells is typically
limited by substrate diffusion, a process that does not or weakly fractionate isotopes. Second, cells maintain
a limited quantity of substrates in the cytoplasm by regulating their substrate absorption and reserves
(Button, 1998). This limited substrate availability prevents the preferential use of light C isotope during
biochemical reactions of cell respiration.
It is well known that the delta $^{13}$C of $CO_2$ emitted from soils shows circadian cycle and seasonal fluctuations
that reaches up to 5‰ (Moyes et al., 2010). However, it is difficult to link these fluctuations to a modification
of metabolic pathways of soil respiration (living respiration versus EXOMET) in response to environmental
changes since numerous other processes can contribute to these fluctuations. Moreover, it is likely that the
EXOMET does not induce much C isotope fractionation in non-sterilized soils since the DOC content is
typically low (**Fig. 3a**) (Liu et al., 2015). Therefore, addition of large amount of DOC is necessary to reveal
the C fractionation induced by the EXOMET in non-sterilized soils.
**Towards a quantification of EXOMET and cellular respiration in living soils**
Our findings support the idea that $CO_2$ emissions from soils are driven by two major oxidative metabolisms:
(1) the well-known respiration of soil biota, (2) an EXOMET carried out by soil stabilized enzymes and soil
minerals and metals. A first quantification of these metabolisms has been made by Maire *et al.* (2013)
suggesting that the EXOMET contributes from 16 to 48 % of soil $CO_2$ emissions. However, Maire *et al.*
(2013) pointed at the need of another method to confirm this substantial contribution of EXOMET. Indeed,
their method can lead to some biases. For instance, the soil irradiation used to block cellular activities and
estimate the EXOMET induces a flush of respiration due to the release of substrates and enzymes from
microbial biomass. This side effect of soil sterilization leads to an overestimation of EXOMET by releasing
enzymes and cofactors in soil.
The difference in C isotope fractionation between EXOMET and cellular respiration offers another method
of quantification of those metabolisms applicable on non-sterilized living soils. The development of this
method first requires a quantification of the isotope fractionation (‰ delta $^{13}$C) and its dependence to DOC
content occurring during cell respiration ($\Delta^{13}C_{cell}$) and EXOMET ($\Delta^{13}C_{EXOMET}$). Our results provide an
example of estimation of $\Delta^{13}C_{EXOMET}$ (**Fig. 3e**), though further studies are needed to verify the genericity of
this estimation in other soils. $\Delta^{13}C_{cell}$ for soil microorganisms can be estimated with cell cultures using soil
inoculum and different substrate concentrations. This quantification allows determining the isotope
composition of $CO_2$ (‰ delta $^{13}C$) released by cell respiration ($\delta^{13}C\text{-}CO_{2cell}$) and EXOMET ($\delta^{13}C\text{-}$
$CO_{2EXOMET}$) in function to DOC content and isotope composition of DOC ($\delta^{13}C\text{-}DOC_{sample}$):
$\delta^{13}C\text{-}CO_{2cell} = \delta^{13}C\text{-}DOC_{sample} - \Delta^{13}C_{cell}$ (1)
$\delta^{13}C\text{-}CO_{2EXOMET} = \delta^{13}C\text{-}DOC - \Delta^{13}C_{EXOMET}$ (2)
with $\Delta^{13}C_{cell}$ and $\Delta^{13}C_{EXOMET}$ are functions of DOC content. Based on our results, $\Delta^{13}C_{EXOMET}$ can be
determined as
$\Delta^{13}C_{EXOMET} = 0.037 \text{ x } [DOC] - 5.495$
where [DOC] is dissolved organic C content ($\mu g$ C $g^{-1}$ soil).
Given that the C isotope fractionation depends on an excess of available substrate, substantial amount of
DOC must be added to the living soil before quantifying EXOMET and cell respiration. After substrate
addition, cellular respiration ($R_{cell}$) and EXOMET ($R_{EXOMET}$) can be separated using the classical isotope
mass balance equations:
$R_{soil} = R_{cell} + R_{EXOMET}$ (3)
$\delta^{13}C\text{-}CO_{2 \text{ soil}} \text{ x } R_{soil} = \delta^{13}C\text{-}CO_{2cell} \text{ x } R_{cell} + \delta^{13}C\text{-}CO_{2EXOMET} \text{ x } R_{EXOMET}$ (4)
where $R_{soil}$ and $\delta^{13}C\text{-}CO_{2 \text{ soil}}$ are respectively the total $CO_2$ emitted by the amended soil ($\mu g$ C-$CO_2$ $kg^{-1}$ soil)
and its isotopic composition (‰ delta $^{13}C$). $R_{soil}$ and $\delta^{13}C\text{-}CO_{2 \text{ soil}}$ must be measured in hours following the
substrate addition before any substantial growth of soil microorganisms which would lead to an over-
estimation of cell respiration. This short-term measurement is also a prerequisite to prevent the microbial
uptake of the heavy C isotope left over by the EXOMET. $\delta^{13}C\text{-}CO_{2cell}$ and $\delta^{13}C\text{-}CO_{2EXOMET}$ must be
estimated in separate experiments as previously described. Therefore, the two unknowns $R_{cell}$ and $R_{EXOMET}$
can be determined by solving the two equations.

**CONCLUSIONS AND IMPLICATIONS**
Collectively, our results tend to sustain the hypothesis through which soil C mineralization is driven by the
well-known microbial mineralization and an EXOMET carried out by soil-stabilized enzymes and by soil
mineral and metal catalysts. These two metabolisms may explain why soil C mineralization is not always
connected to size and composition of the microbial biomass (Kemmitt et al., 2008) and why experimental
reduction of these microbial components has moderate effects on mineralization rate (Griffiths et al., 2001).
Moreover, these two metabolisms should be considered separately when studying effects of environmental
factors on the C cycle because they do not likely obey the same laws and respond differently to
environmental factors. Soil microorganisms have tight physiological constraints comprising specific
environmental conditions (temperature, moisture) and needs in energy and nutrients. The EXOMET is
resistant to extreme conditions (e.g. autoclaving) thanks to soil stabilization of enzymes and depends on
microbial turnover for the supply of respiratory enzymes. These two metabolisms may interact in many
different ways: microbial cells and EXOMET likely compete for available substrates; dying cells are a
source of respiratory enzymes and substrate for the EXOMET etc. Further studies are necessary to better
understand processes at play and predict the relative importance of EXOMET and cell respiration across
ecosystems and climates.

Overall our findings have several implications for biology. They challenge the belief of cell as the minimum structure unit able to organize and achieve cascades of chemical reactions leading to complete oxidation of organic matter. They also suggest that soils have played a key role in the origin of life. Previous studies have shown the role of soil minerals in the concentration and polymerization of amino-acids and nucleic-acids in protein-like molecule during the prebiotic period (Hazen, 2006 ; Bernal, 1949). Our results show that, when all relevant molecules are present, complex biochemical reactions underpinning bioenergetics of life (respiration) can occur spontaneously in the soil. Thus, the first ancestral oxidative metabolisms may have occurred in soil before they were incorporated in the first cell.

## ACKNOWLEDGEMENTS

This work was supported by the project '*Adaptation and responses of organisms and carbon metabolism to climate change*' of the program CPER (French Ministry of Research, CNRS, INRA, Région Auvergne, FEDER) and by the project EXCEED of the program PICS (CNRS). B. Kéraval was supported by a PhD fellowship from the Région Auvergne and the FEDER.

## AUTHOR CONTRIBUTIONS

This work arose from an idea of S.F. and A.C.L.. B.K, S.F, A.C.L, G.A and C.A designed the experiment. B.K and J.C conducted the experiments. B.K analyzed the data. S.F. identified the C isotope fractionation and conceived the model of quantification. B.K, S.F, A.C.L, G.A and C.A co-wrote the paper.

## COMPETING FINANCIAL INTERESTS

The authors declare no conflict of interest

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

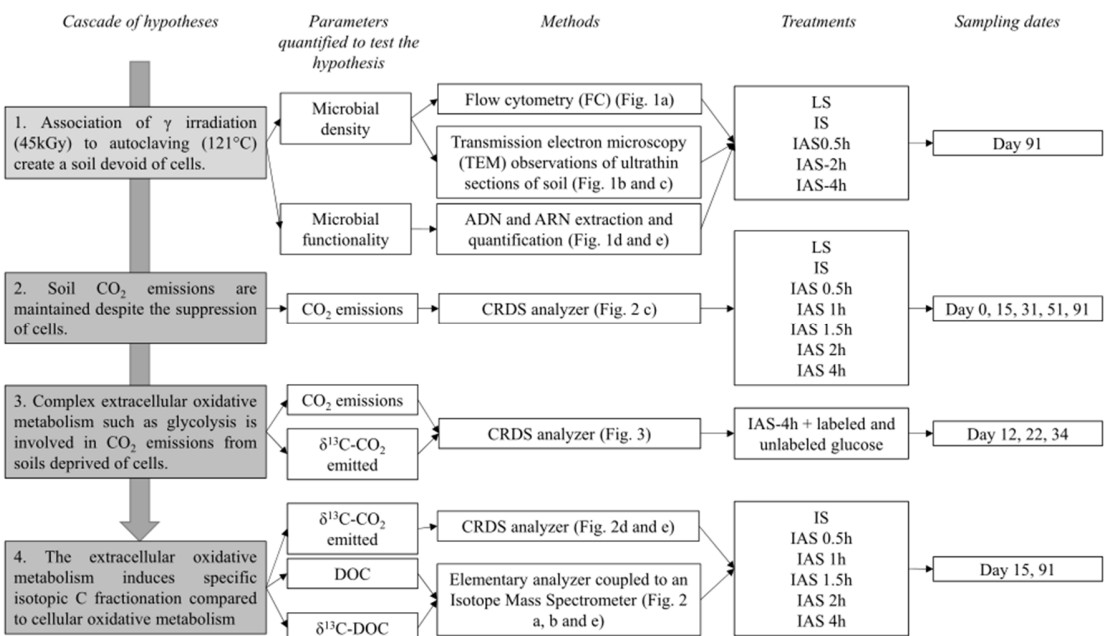


**Figure 1:** General experimental design of the study which include our hypothesis, the parameters, the
methods and the samples (n=3 for each date and treatment studied) used to test those hypotheses.

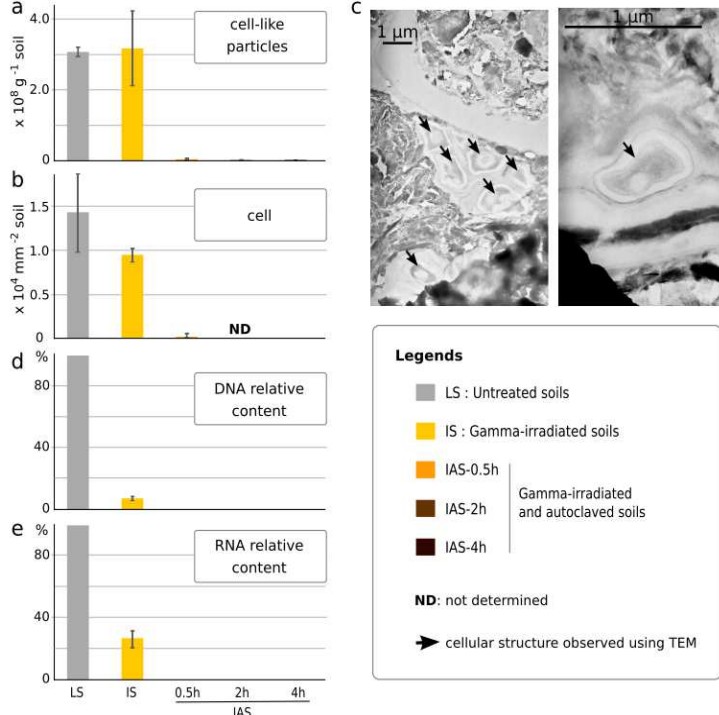

**Figure 2:** Impact of sterilization treatments on cellular density, integrity and functionality.

(a) Cell density enumerated by flow cytometry (FC), (b) cell density and integrity determined by transmission electron microscopy (TEM), (c) TEM photographs of ultrathin sections of soil showing cellular structure in LS, (d) DNA and (e) RNA relative contents in soils (dry mass basis). The percentage of DNA and RNA relative contents was estimated using LS as a reference. Standard deviation was estimated using three replicates per conditions (n=3). LS: Untreated soils, IS: irradiated soils, IAS-t: irradiated and autoclaved soils with 't' referring to the time of autoclaving.

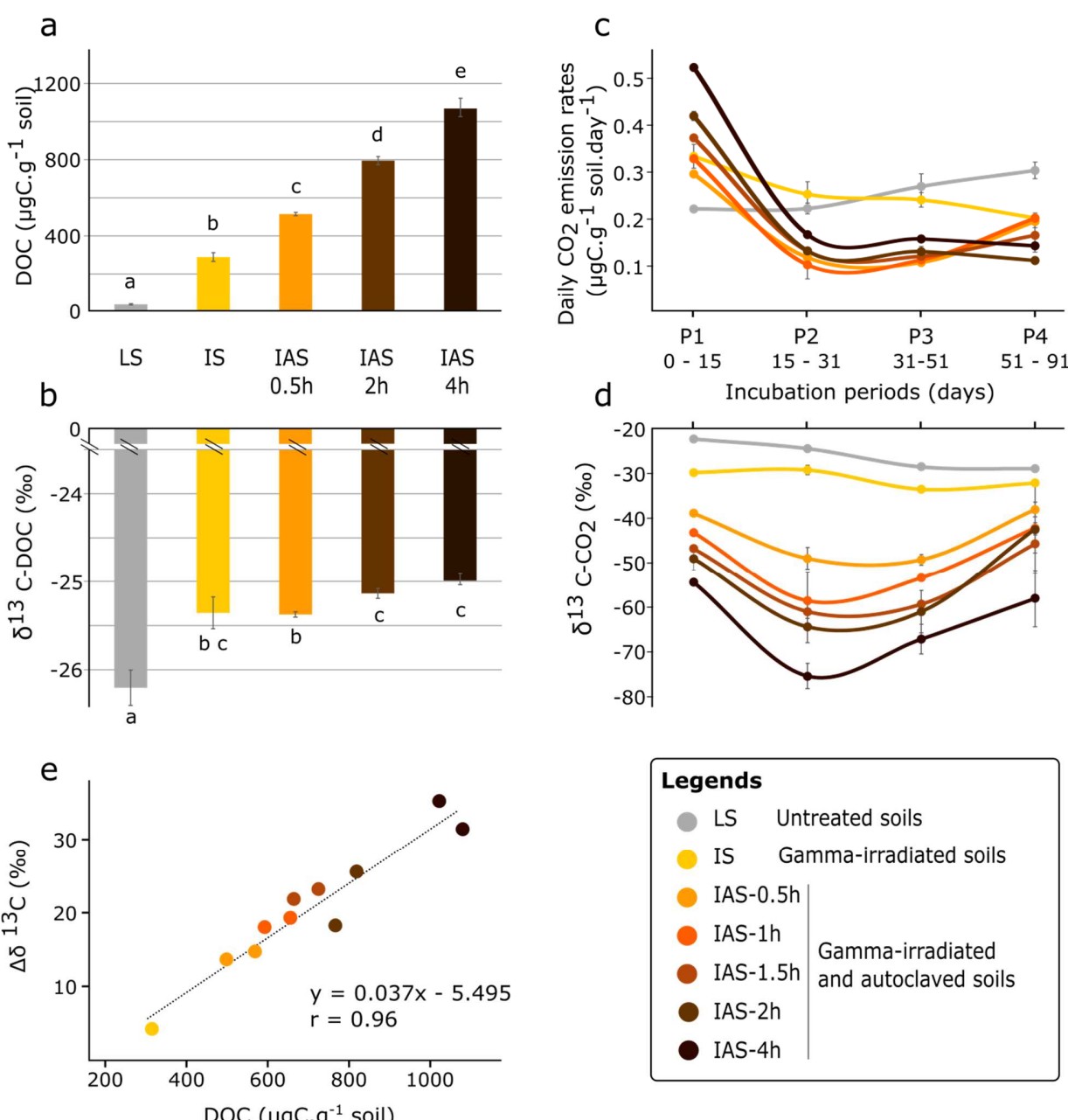

546

**Figure 3:** Content and isotopic composition of dissolved organic carbon (DOC) and of $CO_2$ across time and treatments.

(a) Content and (b) $\delta^{13}C$ of dissolved soil organic carbon content (DOC) at the beginning of incubation, (c) daily $C-CO_2$ emissions rates and (d) $\delta^{13}C$ of $CO_2$ released during four periods of incubation, (e) correlation between the carbon isotope discrimination ($\Delta\delta^{13}C$ in ‰) induced by the extracellular oxidative metabolism (EXOMET) and the DOC content. The correlation was calculated from data of sterilized soil treatments (IS, IAS-0.5h, IAS-1h, IAS-1.5h, IAS-2h, IAS-4h) analyzed at the beginning and the end of incubation. Standard deviation was estimated using three replicates per conditions (n=3). LS: Untreated soils, IS: irradiated soils, IAS-t: irradiated and autoclaved soils with 't' referring to the time of autoclaving.


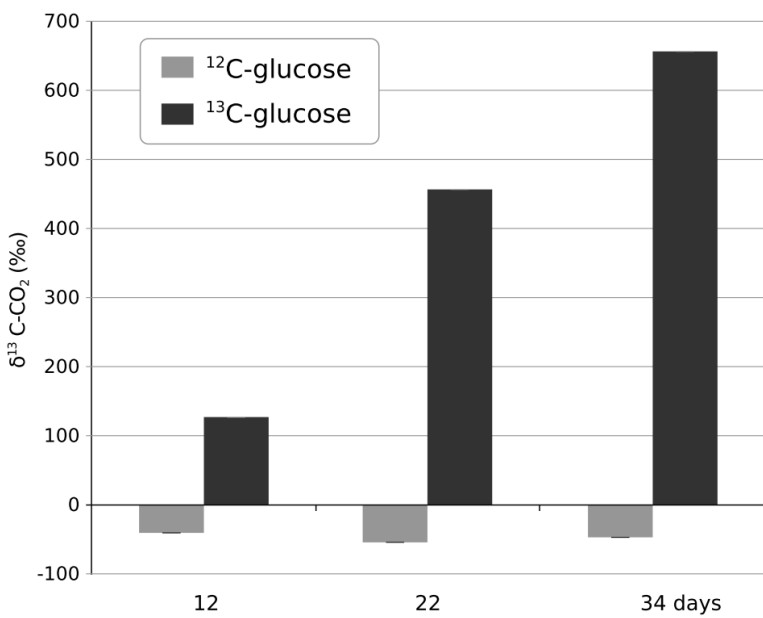


**Figure 4:** Kinetic of the $\delta^{13}$C-CO$_2$ released from an irradiated and autoclaved (4h) soil inoculated with $^{13}$C-
labelled glucose ($^{13}$C-glucose) or with unlabelled glucose ($^{12}$C-glucose) through 34 days of incubation.
Standard deviation was estimated using three replicates per treatments (n=3).