# Peer review of "SOIL CARBON DIOXIDE EMISSIONS CONTROLLED BY AN EXTRACELLULAR 2 OXIDATIVE METABOLISM IDENTIFIABLE BY ITS ISOTOPE SIGNATURE. 3 4 5 Authors: B. Kéraval 1,2,3, A.-C. Lehours1,2, J. Colombet1,2, C. Amblard 1,2, G. Alvarez 3,4, S. Fontaine 3<"

_Biogeosciences, 2015_

## Referee Comment (RC1) · Anonymous Referee #1 · 28 Apr 2016

Review of " Soil carbon dioxide emissions controlled by an extracellular oxidative metabolism identifiable by its isotope signature" by Kéravak and colleagues.

This MS presents interesting data on $CO_2$ released from non-cellular origin in soil. The MS follows up on the previous paper by Maire et al., published in this journal in 2013. The primary goal of this MS is to provide further evidence of the extracellular oxidative metabolism by comparing $CO_2$ released from soil that has undergone different levels of sterilization. An additional goal was to observe whether or not the extracellular metabolic mechanism can break down a relatively complex organic molecule using isotopically labeled glucose.

The MS has improved immensely since the first iteration, especially with the addition

of figure 1 and other clarifications made throughout the text. The methods are appropriate for the questions asked and they have been meticulously carried out. The statistical component is easier to understand, but a few details need to be attended to (see below). The discussion addresses the hypotheses and goals described in the introduction and the author's have pointed out the relevance of the their findings to our current understanding of soil carbon metabolism and how their results can guide future research.

I find the study novel and the results to be very interesting. I think, however, there are a few questions remaining within the results that highlight that the extracellular metabolism is still in the hypothesis phase and that the conclusions the authors draw should reflect this.

My first question concerns the isotope results. From figure 3d, we see CO2 that is very depleted in the heavy isotope (-40 to -55 ‰ at the beginning of the experiment that becomes even more depleted (-50 to -75 ‰, before returning to the beginning values. The authors suggest that this is related to the DOC concentration associated with each autoclave level; however, what is curious to me is that there were no significant differences between the DOC 13C, if the logic is that a low concentration leads to higher fractionation, then we should expect DOC enriched in 13C, but we actually see the opposite (the value in the first bar of fig 4b is about 1‰ depleted relative to the other treatments).

Along this line of reasoning, it seems that a change in the isotopic fractionation should shift linearly only within a treatment, but because there is only a total sample size of 3 and the within treatment DOC concentration variability was small, this cannot be tested. What was done instead, was a comparison across the treatments and I don't entirely agree with this interpretation, simply because the relationship presented in figure 3E is not simply a matter of DOC concentration but also whatever effects (biotic and abiotic) resulted from the treatments.

Thus, I feel the concentration effect as an explanation to the isotopic fractionation effect to be unsatisfying. The precise mechanism seems to still lie within a black box and this study has provided evidence for the extracellular metabolic breakdown of glucose, but much more research remains to fully clarify the processes behind it. Lastly, I think the readers would appreciate it if the authors could put their results in context with what we know already about the isotopic signature of soil respiration. For example, we know that the range extends (normally) from -30 to -23‰ in C3 dominated systems. If the non-cellular breakdown of carbon in soil was significant then shouldn't we expect these values to be much more depleted? Furthermore, how does this theory fit within the diel and seasonal understanding that we have of soil respiration? Perhaps this phenomenon will only be relevant in certain types of soils or climates.

Detailed comments: Page 3 line 28: Aren't most of these enzymes in soils of cellular origin? Page 4 Line 17: probably want to clarify that the sampling was not made continuously. Line 18: maybe reference a biological analog to the "complex cascade of biochemical reactions" to give the reader an idea about what you are describing.

Page 5 Line 2: The beginning of this sentence is confusing – are you trying to make sure that cells were there or were not there.

Section 2.2 I am not aware that picarro sells an injection system for gas samples. Is this a customized unit? Can you also describe how the data were used from the analyzer? For example, normally an injection will have distinct tails as the sample moves through the system, did you take the peak value, integrate, or average over this pulse? Can you also describe the concentration range of your samples and whether or not calibration was necessary?

Page 8 Section 2.9: It is written that the data were tested for normality, but I couldn't find the test results in the results section- is ANOVA justified or should a non-parametric test be used instead?

Page 9 Section 3.12 Were there treatment differences in DOC concentration and the

isotopic signature (not simply between dates as indicated in the text)?

Page 13 line6: I think you mean to say that the "persistence" of emissions or that the emissions were maintained, or something similar.

Page 15 Section 4.4: This section is a fine theoretical example of how to use isotopic information to calculate the contribution of $CO_2$ from the extracellular respiration. The only difficulty is the empirical equation derived from figure 3e. This should be removed for the reasons discussed previously and also to avoid others using the equation under the impression that it might be universal (despite any caveat written in the text).

Figure 1: List the sample size in the figure text. Figure 3a-d: show which treatments are significantly different from each other. In the figure heading list the sample size (n).

---

## Referee Comment (RC2) · Anonymous Referee #2 · 17 May 2016

This excellent study shows the occurence of extracellular respiration in soils and discusses the involved pathways. Even if addressed in earlier works, the question of extracellular or abiotic production of CO2 is of broad interest for the conceptual representation of soil organic carbon mineralization. The study is one of the best conducted on this subject. Even if research has to be continued on this question, these are new concepts and ideas in this study, which are worth being published yet. The initial manuscript has been clearly improved in this new version. I therefore consider the manuscript as acceptable for publication.

Concerning section "4.4. Towards a quantification of EXOMET and cellular respiration in living soils". Results of figure 3e and corresponding equation page 16 that relates

d13C of CO2 to DOC could be explained through two processes of CO2 release by exomet: one involving (almost) no frationation and the other highly fractionating, and probably from carbon derived from extracted/heated organic matter. The linear relationship between d13C and DOC concentration might be as well explained by a proportion of the second process in the CO2 efflux, which is itself correlated with the extraction level of carbon by treatment, as by a reservoir size dependent kinetic expression of the 13C fractionation factor. The proposed method to quantify exomet through 13C signature thus makes sense, but the equation that relates the isotope fractionation to DOC concentration should not be considered as generic.

According to the data, labelled glucose is a source of exomet CO2, but is not the dominant source. The conclusion that exomet can achieve a respiratory-like metabolism doesn't exclude the occurrence in parallel of more partial mineralization processes, e.g. involving methoxy or carboxyls etc. Complete mineralization of complex molecules such as glucose would furthermore lead to smaller isotope fractionation than observed.

---

## Author Comment (AC2) · 1 Aug 2016

Anonymous Referee #2 ()

Referee2: "This excellent study shows the occurence of extracellular respiration in soils and discusses the involved pathways. Even if addressed in earlier works, the question of extracellular or abiotic production of CO2 is of broad interest for the conceptual representation of soil organic carbon mineralization. The study is one of the best conducted on this subject. Even if research has to be continued on this question, these are new concepts and ideas in this study, which are worth being published yet. The initial manuscript has been clearly improved in this new version. I therefore consider the manuscript as acceptable for publication."

[Figure]

Response: We thank the referee for his support and help.

Referee2: "Concerning section "4.4. Towards a quantification of EXOMET and cellular respiration in living soils". Results of figure 3e and corresponding equation page 16 that relates d13C of $CO_2$ to DOC could be explained through two processes of $CO_2$ release by exomet: one involving (almost) no frationation and the other highly fractionating, and probably from carbon derived from extracted/heated organic matter. The linear relationship between d13C and DOC concentration might be as well explained by a proportion of the second process in the $CO_2$ efflux, which is itself correlated with the extraction level of carbon by treatment, as by a reservoir size dependent kinetic expression of the 13C fractionation factor. The proposed method to quantify exomet through 13C signature thus makes sense, but the equation that relates the isotope fractionation to DOC concentration should not be considered as generic."

Response: We completely agree with this point which has also been raised by the first referee. We have modified the text to clarify this limit and suggest studies that could be conducted to overcome these limits (page 14 line 15; page 15 line 17).

Referee2: "According to the data, labelled glucose is a source of exomet $CO_2$, but is not the dominant source. The conclusion that exomet can achieve a respiratory-like metabolism doesn't exclude the occurrence in parallel of more partial mineralization processes, e.g. involving methoxy or carboxyls etc. Complete mineralization of complex molecules such as glucose would furthermore lead to smaller isotope fractionation than observed."

Response: We agree with the idea that there are a few questions remaining within the results that highlight that the extracellular metabolism is still in the hypothesis phase. Therefore, page 16 - line [20-23], our terms were moderated: "Collectively, our results tend to sustain the hypothesis through which soil C mineralization is driven by the well-known microbial mineralization and an EXOMET carried out by soil-stabilized enzymes and by soil mineral and metal catalysts." We have also specified that the causal link

between the magnitude of fractionation and the DOC content is not certain since the correlation emerges from a compilation of results obtained after different sterilization treatments. Further studies should analyze this causal link in experiments where the DOC content is directly manipulated and the change over time of the isotopic composition of DOC is quantified (page 14 line 15; page 15 line 17).

―――――――――――――――――――――

---

## Author Response (AR1)

Benoit Kéraval

CNRS, UMR 6023 - LMGE

Impasse Amélie Murat

63178 Aubière, France

Tel : +33(4) 73 40 74 32

E-mail : benoit.keraval@gmail.com

Clermont Ferrand, 06th June 2016

Dear Editor,

Please find enclosed a revised version of our manuscript "Soil carbon dioxide emissions controlled by an extracellular oxidative metabolism identifiable by its isotope signature".

We would like to thank you and the referees for the constructive comments which did help us to improve the quality of our manuscript.

We hope that this new version will meet the expectations of referees and the standards of *Biogeosciences*.

Best regards,

Benoit Kéraval

**Anonymous Referee #1 ()**

*"This MS presents interesting data on CO2 released from non-cellular origin in soil. The MS follows up on the previous paper by Maire et al., published in this journal in 2013. The primary goal of this MS is to provide further evidence of the extracellular oxidative metabolism by comparing CO2 released from soil that has undergone different levels of sterilization. An additional goal was to observe whether or not the extracellular metabolic mechanism can break down a relatively complex organic molecule using isotopically labeled glucose.*

*The MS has improved immensely since the first iteration, especially with the addition of figure 1 and other clarifications made throughout the text. The methods are appropriate for the questions asked and they have been meticulously carried out. The statistical component is easier to understand, but a few details need to be attended to (see below). The discussion addresses the hypotheses and goals described in the introduction and the author's have pointed out the relevance of their findings to our current understanding of soil carbon metabolism and how their results can guide future research."*

**Response:** We really appreciate the careful analysis of our findings made by the referee. We also thank the referee for the recommendations formulated with the aim to improve our manuscript during the two stages of the reviewing process.

*"I find the study novel and the results to be very interesting. I think, however, there are a few questions remaining within the results that highlight that the extracellular metabolism is still in the hypothesis phase and that the conclusions the authors draw should reflect this."*

**Response:** We agree that EXOMET remains in the hypothesis phase. Therefore, page 16 - line [20-23], our terms were moderated: "Collectively, our results tend to sustain the hypothesis through which soil C mineralization is driven by the well-known microbial mineralization and an EXOMET carried out by soil-stabilized enzymes and by soil mineral and metal catalysts."

*"My first question concerns the isotope results. From figure 3d, we see CO2 that is very depleted in the heavy isotope (-40 to -55 ‰ at the beginning of the experiment that becomes even more depleted (-50 to -75 ‰, before returning to the beginning values. The authors suggest that this is related to the DOC concentration associated with each autoclave level; however, what is curious to me is that there were no significant differences between the DOC 13C, if the logic is that a low concentration leads to higher fractionation, then we should expect DOC enriched in 13C, but we actually see the opposite (the value in the first bar of fig 4b is about 1‰ depleted relative to the other treatments)."*

**Response:** In fact, figure 3b presents the delta [13]C of DOC at the beginning of experiment, that is, before the EXOMET might have changed the delta [13]C of DOC due to its isotopic discrimination activities (this is specified in the figure caption). Therefore, it is not surprising to see any important difference between treatments. However, we agree that the causal link between the magnitude of fractionation and the DOC content is not certain and deserves other studies. We added two sentences (page 14 line 15, page 15 line 17) conveying this message.

*"Along this line of reasoning, it seems that a change in the isotopic fractionation should shift linearly only within a treatment, but because there is only a total sample size of 3 and the within treatment DOC concentration variability was small, this cannot be tested. What was done instead, was a comparison across the treatments and I don't entirely agree with this interpretation, simply because the relationship presented in figure 3E is not simply a matter of DOC concentration but also whatever effects (biotic and abiotic) resulted from the treatments.*

*Thus, I feel the concentration effect as an explanation to the isotopic fractionation effect to be unsatisfying. The precise mechanism seems to still lie within a black box and this study has provided evidence for the extracellular metabolic breakdown of glucose, but much more research remains to fully clarify the processes behind it."*

**Response:** As explained above we agree with these ideas and we have added two sentences acknowledging the limits of our study and explaining what can be done to progress.

*Lastly, I think the readers would appreciate it if the authors could put their results in context with what we know already about the isotopic signature of soil respiration. For example, we know that the range extends (normally) from -30 to -23‰ in C3 dominated systems. If the non-cellular breakdown of carbon in soil was significant then shouldn't we expect these values to be much more depleted? Furthermore, how does this theory fit within the diel and seasonal understanding that we have of soil respiration? Perhaps this phenomenon will only be relevant in certain types of soils or climates."*

**Response:** We have added the following paragraph to discuss this idea:

"It is well known that the delta $^{13}$C of $CO_2$ emitted from soils shows circadian cycle and seasonal fluctuations that reaches up to 5‰ (Moyes et al., 2010). However, it is difficult to link these fluctuations to a modification of metabolic pathways of soil respiration (living respiration versus EXOMET) in response to environmental changes since numerous other processes can contribute to these fluctuations. Moreover, it is likely that the EXOMET does not induce much C isotope fractionation in non-sterilized soils since the DOC content is typically low (**Fig. 3a**) (Liu et al., 2015). Therefore, addition of large amount of DOC is necessary to reveal the C fractionation induced by the EXOMET in non-sterilized soils."

*Detailed comments:*

*"Page 3 line 28: Aren't most of these enzymes in soils of cellular origin?"*

**Response:** To avoid confusion we changed the sentence by: "(i) suggest that $CO_2$ emissions from soils are not only dependent to the bio-physicochemical environment provided by the cells".

*"Page 4 Line 17: probably want to clarify that the sampling was not made continuously."*

**Response:** We changed the sentence Page 4 Line 17 by: "The production and the isotope composition
($\delta^{13}C$) of $CO_2$ were monitored in sterilized and non-sterilized soils over 4 periods through 91 days of
incubation.".

*"Line 18: maybe reference a biological analog to the "complex cascade of biochemical reactions"*
*to give the reader an idea about what you are describing."*

**Response:** We changed the sentence Page 4 Line 18 by: "We also tested whether the EXOMET in
sterilized soils can carry out complex cascade of biochemical reactions (e.g. an equivalent of glycolysis and
Krebs cycle) by incorporating $^{13}C$- labelled glucose and by quantifying emissions of $^{13}C$-$CO_2$ **(Fig 1)**."

*"Page 5 Line 2: The beginning of this sentence is confusing – are you trying to make sure that cells*
*were there or were not there."*

**Response:** We changed the sentence Page 5 Line 2 by: "To demonstrate the absence of viable cells in
soil after irradiation, …"

*"Section 2.2 I am not aware that picarro sells an injection system for gas samples. Is this a*
*customized unit? Can you also describe how the data were used from the analyzer? For example,*
*normally an injection will have distinct tails as the sample moves through the system, did you take*
*the peak value, integrate, or average over this pulse? Can you also describe the concentration*
*range of your samples and whether or not calibration was necessary?"*

**Response:** We improved this paragraph following your recommendations: "The amount and isotope
composition ($\delta$ $^{13}C$) of $CO_2$ accumulated in flasks during the incubation period were quantified using a
cavity ring down spectrometer analyser coupled to a small sample injection module (Picarro 2101-i analyser
coupled to the SSIM, Picarro Inc., Santa Clara, CA, USA). A volume of 20 ml of gas was sampled by the
analyser. The $CO_2$ concentration in gas samples ranged from 300 to 2000 ppm of $CO_2$ in accordance with
the operating range of the analyser. The $CO_2$ concentrations and delta $^{13}C$ of gas samples were measured at
a frequency of 30 $mn^{-1}$ during 10 mn. Value provided by the analyser is the integrated value during these 10
mn of measurement. A reference gas with a known concentration of $CO_2$ and delta $^{13}C$ was injected between
samples. For each period of incubation, the cumulated amount of $CO_2$ was divided by the duration of the
period (in days) to estimate the mean daily $CO_2$ emission rate."

*Page 8 Section 2.9: It is written that the data were tested for normality, but I couldn't find the test*
*results in the results section- is ANOVA justified or should a non-parametric test be used instead?"*

**Response:** We have indicated the p-values ranges that we used to test the normal distribution of
our values and the equality of the variances: Page 8 Line 20 "Normality was tested using the Shapiro-
Wilk test ($p>0.05$). Equality of variances were tested with a Leven's Test ($p<0.05$).".

*"Page 9 Section 3.12 Were there treatment differences in DOC concentration and the isotopic*
*signature (not simply between dates as indicated in the text)."*

**Response:** There is only one date of measurement, at the beginning of the experiment. We have
slightly modified this paragraph in order to clarify the presentation of results: "Both γ-irradiations
and autoclaving modified the soil chemistry as revealed by the analysis of the aqueous phase at the beginning of the experiment. The aqueous phase contained much more DOC in irradiated soil than in untreated soil $(37\pm3$ µg $C.g^{-1}$ to $303\pm17$ µg $C.g^{-1}$ in LS and IS, respectively (**Fig. 3a**)."

*"Page 13 line6: I think you mean to say that the "persistence" of emissions or that the emissions were maintained, or something similar."*

**Response:** You are right. We have changed the sentence by: "Moreover, Blankinship *et al.* (**Blankinship et al., 2014**) have shown that the persistence of soil $CO_2$ emissions after microbial biomass suppression (or at least reduction) is not specific to irradiated soil but also occurs with other methods of sterilization such as chloroform fumigation and autoclaving."

*"Page 15 Section 4.4: This section is a fine theoretical example of how to use isotopic information to calculate the contribution of CO2 from the extracellular respiration. The only difficulty is the empirical equation derived from figure 3e. This should be removed for the reasons discussed previously and also to avoid others using the equation under the impression that it might be universal (despite any caveat written in the text)."*

**Response:** In fact, we wanted to present this equation as an example of how this fractionation coefficient can be calculated. We agree with you that this coefficient can vary across soils and should not be viewed as a generic coefficient (at least at this step of knowledge). We have modified the paragraph to clarify this point.

*"Figure 1: List the sample size in the figure text. Figure 3a-d: show which treatments are significantly different from each other. In the figure heading list the sample size (n)."*

**Response:** Following your recommendations, we have listed the sample size (n=3) in the text of figure 1, 2, 3, 4. We have also showed the differences significance between treatment in figure 3a-b. However, we did not show those last results in figure 3c-d in order to improve the readability of those figures. Standard deviations represent sufficient statistical tools which allow to illustrate the results and the messages described in paragraph 3.1.3.

**Anonymous Referee #2 ()**

*"This excellent study shows the occurence of extracellular respiration in soils and discusses the involved pathways. Even if addressed in earlier works, the question of extracellular or abiotic production of CO2 is of broad interest for the conceptual representation of soil organic carbon mineralization. The study is one of the best conducted on this subject. Even if research has to be continued on this question, these are new concepts and ideas in this study, which are worth being published yet. The initial manuscript has been clearly improved in this new version. I therefore consider the manuscript as acceptable for publication."*

**Response:** We thank the referee for his support and help.

*"Concerning section "4.4. Towards a quantification of EXOMET and cellular respiration in living*
*soils". Results of figure 3e and corresponding equation page 16 that relates d13C of CO2 to DOC*
*could be explained through two processes of CO2 release by exomet: one involving (almost) no*
*frationation and the other highly fractionating, and probably from carbon derived from*
*extracted/heated organic matter. The linear relationship between d13C and DOC concentration*
*might be as well explained by a proportion of the second process in the CO2 efflux, which is itself*
*correlated with the extraction level of carbon by treatment, as by a reservoir size dependent kinetic*
*expression of the 13C fractionation factor. The proposed method to quantify exomet through 13C*
*signature thus makes sense, but the equation that relates the isotope fractionation to DOC*
*concentration should not be considered as generic."*

**Response:** We completely agree with this point which has also been raised by the first referee. We
have modified the text to clarify this limit and suggest studies that could be conducted to overcome
these limits (page 14 line 15; page 15 line 17).

*"According to the data, labelled glucose is a source of exomet CO2, but is not the dominant source.*
*The conclusion that exomet can achieve a respiratory-like metabolism doesn't exclude the*
*occurrence in parallel of more partial mineralization processes, e.g. involving methoxy or*
*carboxyls etc. Complete mineralization of complex molecules such as glucose would furthermore*
*lead to smaller isotope fractionation than observed."*

**Response:** We agree with the idea that there are a few questions remaining within the results that
highlight that the extracellular metabolism is still in the hypothesis phase. Therefore, page 16 - line

[revised manuscript text omitted]

---

## Author Response (AR2)

Comments to the Author:

The authors have addressed the concerns of the two reviewers and the manuscript is now nearly ready for publication. My main question that remains is how the proportion of DOC oxidized to CO2 changes with treatment (i.e., divide CO2 production rate by DOC content). The DOC increases with duration of autoclaving but does the CO2 emission rate increase proportionally? This could be one more useful piece of information to help interpret the purported isotopic fractionation effects and their dependence on DOC concentration. See my comments on Fig 3 and suggestions for including this in the discussion section.

Response: We thank the editor for his careful reading of new manuscript and responses to referee's comments. We also appreciate his suggestion of a new figure presenting the proportion of DOC oxidized to CO2. This figure has been inserted as supplementary material. The percentage of DOC oxidized to CO2 was very low in all sterilized soils and decreased with the duration of autoclaving (and with the increase of DOC concentration). This result supports the idea that the EXOMET and its fractionation were quantified with an excess of substrate (this information has been added in the manuscript, see below). However, this result does not change the dependence of isotopic fractionation to DOC concentration.

Specific comments:

Line 31: and line 379: remove "to" so it reads "do not likely obey the same…".
Response: The sentence has been corrected.

Line 56: and line 291: change "glucose in…" to "glucose to…" and line 292, "oxidized to".
Response: The sentence has been corrected.

Lines 109-110: I don't think "mn" is an accepted abbreviation for minutes, rather use "min". Also 30 mn^-1 would more commonly be 0.5 Hz (one measurement per 2 seconds).
Response: The abbreviation has been corrected.

Line 130: Is FC flow cytometry? Please spell out on first use.
Response: The abbreviation is now defined.

Line 219-221 etc: Please remove the unnecessary abbreviation DER.
Response: The paragraph has been modified following your recommendations.

Line 395: change to "before they were incorporated in the first cell."
Response: The sentence has been corrected.

Figure 1: This is a helpful figure, however it needs to be edited to change the figure numbers within it, because perhaps it was prepared as a supplemental figure initially; change Fig 1a to 2a, etc.
Response: The figure has been modified with the correct figure numbers.

Figure 3: What are the units on 3c? This figure would look really different if presented per unit of DOC.

Response: Units of y-axis of figure 3c are $\mu g\ C\ g^{-1}$ soil day$^{-1}$. Sorry they disappeared during the edition of figures. Figure 3 presents the daily $CO_2$ emission rates of treated soils for the four periods of incubation. We cannot calculate the ratios respired/DOC for each period of incubation because we only measured the DOC content at the beginning of incubation. Thus, it is not possible to present Figure 3c with respiration rates expressed per unit of DOC.

Nevertheless, we calculated the % of initial DOC oxidized to $CO_2$ during the incubation (Total emitted $CO_2$/initial DOC), see below.

Figure 4: The caption says 32 days but bar graph label says 34 days.

Response: The caption has been modified.

Line 229: Has a mass balance been done on the DOC and $CO_2$? Probably the amount of $CO_2$ release was too small to make a change in the large DOC pool, but it would be useful to know what % of the DOC was cumulatively respired (similar to your estimate for the % of glucose input on lines 241-242).

Response: We calculated the % of the initial DOC oxidized to $CO_2$ and built a figure presented as supplementary material. Overall, this % was low for all sterilized soils (< 7.2%) and decreased with the duration of autoclaving (from 2.9 to 1.8% for IAS 0.5H and IAS 4H respectively). This information has been added lines 227-229.

Lines 268-284: Consider discussing the amount of $CO_2$ respired per unit of DOC in this section (see comments on Fig 3). Of course, this will be highest in LS and lower in the others, but from Figure 3a and 3c it's not clear how the other treatments will affect the efficiency of C respiration. How does this change the argument about the fractionation effect of EXOMET? Or, this could be discussed in the section on lines 304-318.

Response: After some discussions we came to the conclusion it was not relevant to include this result (% of respired DOC) when we present the body of evidence of EXOMET (Lines 268-284 of old version of manuscript). Instead of supporting the main message it would bring some confusions because 1°) there are already many hypotheses considered and discussed, and 2°) this result is not useful to exclude or support one of presented hypotheses.

However we find highly relevant to use this result in the discussion section dedicated to the isotopic fractionation of EXOMET (Lines 322-323). Indeed, the low percentage of DOC respired indicated that the EXOMET and its fractionation was quantified with an excess of substrate, which is the condition sine qua non for detecting a fractionation process.